# Extensive Phenotypic Characterization of T Cells Infiltrating Liver Metastasis from Colorectal Cancer: A Potential Role in Precision Medicine

**DOI:** 10.3390/cancers14246069

**Published:** 2022-12-09

**Authors:** Gabriela Sampaio-Ribeiro, Ana Ruivo, Ana Silva, Ana Lúcia Santos, Rui Caetano Oliveira, Paula Laranjeira, João Gama, Maria Augusta Cipriano, José Guilherme Tralhão, Artur Paiva

**Affiliations:** 1Flow Cytometry Unit, Clinical Pathology Department, Centro Hospitalar e Universitário de Coimbra EPE, 3000-075 Coimbra, Portugal; 2Institute for Clinical and Biomedical Research (iCBR), Faculty of Medicine, University of Coimbra, 3000-548 Coimbra, Portugal; 3Center for Innovative Biomedicine and Biotechnology (CIBB), University of Coimbra, 3000-548 Coimbra, Portugal; 4Surgery Department, Centro Hospitalar e Universitário de Coimbra, 3000-075 Coimbra, Portugal; 5Faculty of Medicine, University of Coimbra, 3000-548 Coimbra, Portugal; 6Germano de Sousa—Centro de Diagnóstico Histopatológico CEDAP, 3000-377 Coimbra, Portugal; 7Centre of Investigation on Genetics and Oncobiology (CIMAGO), Faculty of Medicine, University of Coimbra, 3000-548 Coimbra, Portugal; 8Clinical and Academic Center of Coimbra (CACC), 3000-075 Coimbra, Portugal; 9Center for Neuroscience and Cell Biology (CNC), Faculty of Medicine, Polo 1, 1st Floor, University of Coimbra, 3004-504 Coimbra, Portugal; 10Pathology Department, Centro Hospitalar e Universitário de Coimbra, 3000-075 Coimbra, Portugal; 11Ciências Biomeédicas Laboratoriais, ESTESC-Coimbra Health School, Instituto Politeécnico de Coimbra, 3046-854 Coimbra, Portugal

**Keywords:** colorectal cancer, liver metastasis, T cells, therapy, personalized medicine

## Abstract

**Simple Summary:**

Cancer cell progression and establishment at the secondary site are influenced by the immune system, meaning that its characterization is crucial to target the metastatic process in oncological patients. As liver metastasis is the principal cause of death in colorectal cancer (CRC) patients, discovering which functional T-cell subsets are present will possibly unravel new targets in the tumor microenvironment. In this study, we characterized several T-cell populations by flow cytometry in non-tumor and tumor samples of CRC liver metastasis, both subdivided according to their growth pattern into desmoplastic and non-desmoplastic. A tendency toward an immunosuppressive microenvironment has been noted, but we also found a significant increase in follicular cytotoxic T cells. Further studies are crucial for fully characterizing the molecular mechanisms involved in CRC liver metastasis formation. Our findings, however, identify new prospective targets to treat these patients.

**Abstract:**

Colorectal cancer (CRC) is one of the most common cancers worldwide, with liver metastasis being its main cause of death. This study harvested fresh biological material from non-tumor and tumor tissue from 47 patients with CRC liver metastasis after surgery, followed by mechanical cellular extraction and stain-lyse-wash direct immunofluorescence technique. Here, 60 different T-cell populations were characterized by flow cytometry. Tumor samples were also subdivided according to their growth pattern into desmoplastic and non-desmoplastic. When we compared tumor versus non-tumor samples, we observed a significantly lower percentage of T-lymphocyte infiltration in the tumor in which the CD4^+^ T-cell density increased compared to the CD8^+^ T cells. T regulatory cells also increased within the tumor, even with an activated phenotype (HLA-DR^+^). A higher percentage of IL-17-producing cells was present in tumor samples and correlated with the metastasis size. In contrast, we also observed a significant increase in CD8^+^ follicular-like T cells (CD185^+^), suggesting a cytotoxic response to cancer cells. Additionally, most infiltrated T cells exhibit an intermediate activation phenotype (CD25^+^). In conclusion, our results revealed potential new targets and prognostic biomarkers that could take part in an algorithm for personalized medicine approaches improving CRC patients’ outcomes.

## 1. Introduction

Colorectal cancer (CRC) is one of the three most common cancers and the second most common cause of cancer-related deaths worldwide [1]. Metastasis remains the major cause of death in CRC that predominantly metastasizes to the liver [2]. Approximately 25% of patients have already developed metastasis at diagnosis, and up to 40–50% will develop liver metastasis after tumor resection, reducing the five-year survival rate to 10–20% compared to 80–90% of patients with localized disease [3,4,5]. These numbers remain unchanged over the years, demonstrating the need for new studies to create new therapeutic approaches.

CRC liver metastasis exhibits two distinct histological growth patterns depending on the interface between the tumor and the surrounding parenchyma, designated as desmoplastic type or non-desmoplastic type [6]. The desmoplastic pattern is characterized by the formation of a band of desmoplastic stroma, allowing a clear demarcation between the tumor and the liver parenchyma. The non-desmoplastic pattern is often subdivided into a replacement pattern, characterized by cancer cell infiltration in the liver parenchyma without disrupting the tissue, and a pushing pattern characterized by the compression of the surrounding hepatocytes by cancer cells—both are characterized by a poor outcome [7]. Additionally, multiple studies suggest a distinct distribution of the immune cells in the different histological growth patterns [8,9].

The immunological microenvironment is essential for the spread of CRC to the liver [10,11]. Therefore, a suppressive immunologic microenvironment has a tumor-promoting function in metastases formation, which is related to tumor-associated macrophages and T regulatory cells (Treg) [10]. On the other hand, tumor-infiltrating lymphocytes are highly heterogeneous in terms of their cell-type composition and functional properties and are associated with improved survival in CRC [12]. 

T regulatory cells (Tregs) that could be identified based on the specific phenotypes CD25^+/bright^ and CD127^−/dim^ are known as powerful immunosuppressive mediators essential to the resolution of inflammation and restoration of the immune response [13]. There is mounting evidence showing that an accumulation of Tregs in a variety of carcinomas is linked to a poor prognosis [14]. Additionally, Tregs control several molecules, including IL-2 and IL-10, and promote tolerance in the tumor microenvironment (TME), making it a possible target for cancer therapy [15].

Moreover, CD4 T helper cells (Th) exert a key role in antitumor immunity. Indeed, the involvement of Th17 cells in inflammatory diseases and CRC is nowadays widely accepted [16]. Studies have shown that chronic Th17-dependent inflammation of the intestine favors angiogenesis and immune subversion and promotes tumor outgrowth [17]. In contrast, increased expression of Th1-cluster genes in resected CRC specimens is associated with improved disease-free survival [18,19]. These cells improve the antitumoral immune response and the effector capabilities of T cells. CD8^+^ T cells can also contribute to tumor outgrowth favoring inflammation through the release of IL-17, designed as Tc17 cells [20]. However, the presence and effects of Th17 and Tc17 cells in the context of CRC liver metastasis are still unknown.

Additionally, follicular T cells are also detected in the TME. These cells, which are crucial for the establishment of humoral immunity [21], are characterized by the expression of CXC-chemokine receptor 5 (CXCR5), or CD185 [22]. Distinct subpopulations of follicular T cells can be identified based on the expression of CXCR3, CCR6, key transcription factors, and cytokines; in addition, these subpopulations display distinct abilities to provide help to naïve and memory B cells. Following activation, follicular T cells upregulate activation markers, such as the co-inhibitory receptor programmed cell death 1 (PD1), inducible T-cell co-stimulator (ICOS), and CD38, express signature transcription factors, such as B cell lymphoma 6 (BCL-6) and increased IL-21 and IL-10 production, which can support their role as immune mediators in CRC [23]. 

Taking these factors into account, this study aimed to identify approximately 60 different T-cell subpopulations characterized by flow cytometry in tissue samples from patients with CRC liver metastasis, which are further subdivided into non-tumor and tumor samples with non-desmoplastic or desmoplastic growth patterns, in order to characterize the TME and clarify new potential targets to inhibit CRC liver metastasis progression.

## 2. Materials and Methods

### 2.1. Participants and Sample Collection

Non-tumor and tumor samples from 47 individuals with CRC liver metastasis (patients’ demographic and clinical characteristics presented in Table 1) were collected at the Centro Hospitalar e Universitário de Coimbra (CHUC). The study was approved by the institution’s ethics committee (number CHUC-127-19), and all participants gave their signed and informed consent before entering the study. Fresh biological material was harvested in the first 30 min after surgical resection by an experienced pathologist. The surgical specimen was carefully sectioned, and circa 3–5 mm^3^ of tumoral and non-tumoral tissue was collected at the tumor/liver interface, as shown in Figure 1a.

### 2.2. Histological Analysis

Regarding the analysis of liver metastases samples from CRC, the growth pattern in the tumor periphery of the liver metastasis was evaluated as described by Van den Eynden et al. [24]: desmoplastic growth pattern when the metastases present a border of connective tissue between the tissue and liver parenchyma; expansive growth pattern when liver cell plaques at the interface are compressed by the tumor; and replacement growth pattern when infiltrating tumor cells replace hepatocytes in liver cell plaques. The pattern must represent more than 75% of the interface. In cases where there are two different patterns, each presenting more than 25% of the invasive front of the tumor, the growth pattern is called mixed [25]. The growth pattern was further dichotomized as desmoplastic versus non-desmoplastic, reflecting its biological behavior [6].

All slides were observed using a light microscope—Nikon Eclipse 50i, and the images were obtained using a Nikon-Digital Sight DS-Fi1 camera (Figure 1b–d).

### 2.3. Flow Cytometry Characterization of T Cells

#### 2.3.1. Staining Protocol

Non-tumor and tumor liver samples of CRC metastases were submerged in phosphate-buffered saline (PBS, Gibco, Life Technologies, Paisley, UK) and repeatedly injected with PBS for the release of the maximum number of cells into the supernatant. After this procedure, the suspensions were collected in a 15 mL Falcon tube and centrifuged at 500× *g* for 5 min. The supernatant was discarded, and the pellet was resuspended in 1 mL of PBS. 

The cellular suspension of each sample was stained for cell surface markers using a stain-lyse-wash direct immunofluorescence technique. An eight-color monoclonal antibody combination panel was used to identify the different T cells’ subpopulations of interest. Details of the antibodies’ characteristics are described in Table 2. The monoclonal antibodies were added to 300 μL of the cell suspension and incubated for 10 min in the dark, at room temperature. Next, 2 mL of FACSLysing solution (BD, Becton Dickinson Biosciences, San Jose, CA, USA) was added, and after 10 min of incubation in the dark, at room temperature, the samples were centrifuged at 500 g for 4 min. The supernatant was discarded, and the cell pellet was washed in 1 mL of PBS and then resuspended in 500 μL of PBS.

#### 2.3.2. Data Acquisition and Analysis

Data were acquired in a FACSCanto II (BD) flow cytometer equipped with the FACSDiva software (v6.1.2; BD, San Jose, CA, USA). Note that all results were acquired according to the correct compensation controls following all demanded recommendations by the EuroFlow consortium [26]. For data analysis, the Infinicyt™ software, V.1.8 (Cytognos SL, Salamanca, Spain), was used. The gating strategy used for the identification of T-cell subpopulations and phenotypic characteristics is represented in Figure 2.

### 2.4. Statistical Analysis

All results are expressed as mean ± standard error of the mean (SEM). Mann–Whitney non-parametric tests and Pearson tests were applied to determine the significance of the differences between the different experimental conditions, as appropriate, using GraphPad Prism Software version (GraphPad Software, San Diego, CA, USA). Values were considered statistically significant if *p* < 0.05.

## 3. Results

### 3.1. The Liver Niche Decreases the Migration of Pro-Inflammatory Cells to the CRC Metastasis Location

To characterize the immune microenvironment of liver metastasis of CRC, samples from non-tumoral and tumor liver tissue were used for a flow cytometry analysis. 

Based on the T cell markers used in this study, we identified 60 different T cell populations in each sample of CRC liver metastasis. The non-tumor and tumor samples were compared to evaluate differences in the frequency of T cell populations in both conditions. In a secondary analysis, the tumor samples were subdivided into two groups, according to the histological characterization previously performed based on the tumor growth pattern: tumor samples with desmoplastic growth pattern and non-desmoplastic, which includes the expansive and replacement patterns. 

According to our data, the overall percentage of T cells significantly decreases in tumor samples compared to non-tumor samples (*p* < 0.0001) (Table 3), even when we remove from the total cellularity the non-hematopoietic cells. Of note, the percentage of CD8^+^, CD4^+^CD8^+^ and γδ^+^ cells also decrease in tumor samples (*p* < 0.0001). On the other hand, we obtained a significant increase in the CD4^+^ T cells in the same conditions (*p* < 0.0001). These results reveal a decreasing migration of cells with cytotoxic ability into the TME. Interestingly, in the analysis of tumor samples with non-desmoplastic and desmoplastic growth patterns, the T cells with cytotoxic capacity were increased in the desmoplastic samples (CD8^+^; *p* < 0.05).

Additionally, the identification of CD56^+^ cells was also assessed to verify the existence of a NK-like phenotype in CD4^+^ and CD8^+^ T cells (Table 3). This analysis revealed, in tumor samples, a significant decrease in the frequency of both populations expressing CD56 (*p* < 0.0001) compared to non-tumor samples, which corroborates the results previously obtained, indicating a lack of cytotoxic T cells in the TME. Again, an increased frequency of CD4^+^ and CD8^+^ T cells expressing CD56 was observed in tumor samples with a desmoplastic growth pattern (*p* < 0.0001).

No other differences regarding the histopathological growth patterns were found in this study. 

### 3.2. T Regulatory and Follicular-like Cells Are Increased in Tumor Samples of CRC Liver Metastasis

The presence of T regulatory cells (Tregs) was significantly increased in tumor samples in three T-cell subpopulations: CD4^+^ (*p* < 0.0001), CD8^+^ (*p* < 0.0001) and CD4^+^CD8^+^ (*p* < 0.01). (Figure 3a–c). The increased percentage of Treg cells suggests that the TME is prompt to be more anti-inflammatory, reinforcing the resistance of tumor cells and promoting their growth. Within this population, the follicular-like phenotype in Tregs was also evaluated but only in CD4^+^ T cells due to the lower number of events obtained in the CD8^+^ and CD4^+^CD8^+^ Tregs cells populations. The follicular cells were observed by the expression of CD185 and their activation status by the expression of HLA-DR. We observed an increase of activated non-follicular Treg cells in tumor samples, suggesting that these cells are actively performing their function (*p* < 0.0001) (Figure 3d).

Additionally, the other follicular-like T cells were also evaluated. An increased frequency of follicular CD8^+^ and CD4^−^CD8^−^ T cells was observed in tumor samples (Figure 3f,h), as well as a decreased percentage of activated follicular CD4^+^ and γδ^+^ T cells (Figure 3e,i). No significant differences were observed in follicular CD4^+^CD8^+^ T cells (Figure 3g). 

### 3.3. Th17 Cells Are Attracted to the Tumor Microenvironment According to the Tumor Size

The T helper cells are also a focus of several studies regarding CRC metastatic progression. On this note, we accessed the expression of surface markers typically associated with Th1 and Th17 functional phenotypes to verify their contribution to the metastatic liver microenvironment.

We evaluated the expression of CCR5^+^ CXCR3^+^, CCR5^−^ CXCR3^+^, CCR5^+^ CXCR3^−^, CCR5^−^ CXCR3^−^, and CCR6^+^ in CD4^+^ and CD8^+^ T-cell subsets. In tumor samples, a significant decrease percentage of CCR5^+^ CXCR3^−^ in both CD4^+^ and CD8^+^ T cells was observed (*p* < 0.05 and *p* < 0.001, respectively). Moreover, a significant increase of CD4^+^ T cells expressing CCR6 (*p* < 0.05) and, in a lower extension, on CD8^+^ T cells (Figure 4a,c) were found. These results show a tendency to a decreased infiltration of T cells with a Th1-like phenotype and an increase of Th17 cells in the TME compared to the adjacent non-tumor tissue.

Additionally, we also looked for Th17/Th1 plasticity based on the concomitant expression of CCR6 and CCR5 and/or CXCR3. This analysis showed, in tumor samples, an increased percentage of CCR6^+^ CD8^+^ T cells expressing both CCR5 and CXCR3 (*p* < 0.01) and a decrease of Th17 and Tc17 cells expressing only CCR5 (*p* < 0.01 and *p* < 0.0001, respectively) and an increase of Tc17 cells expressing only CXCR3 (*p* < 0.0001). These results also revealed an increased frequency of Tc17 negative for CCR5 and CXCR3- (*p* < 0.05), somehow more committed to producing Th17-type cytokines (Figure 4b,d). 

In this evaluation, we could also verify that the tumor size is significantly correlated with the presence of Th17/Th1 (CD4^+^ CCR6^+^ CCR5^+^ CXCR3^−^) and Th17 cells (CD4^+^ CCR6^+^ CCR5^−^ CXCR3^−^) cells in the TME (Figure 4e,f). As the size of the tumor increases, the percentage of these T cells also increases, revealing their augmented presence within the TME. 

### 3.4. T Cells Infiltrating CRC Liver Metastasis Present an Intermediate Activation Phenotype

The activation profile of T cells, after removing Tregs cells from all T-cell compartments, was evaluated in the T cells’ major subpopulations (CD4^+^, CD8^+^, CD4^+^ CD8^+^, CD4^−^ CD8^−^, and γδ^+^) based on the expression of CD25, an intermediate activation marker, and HLA-DR, a late activation marker.

The results have shown a significant increase in CD25^+^ T cells in the tumor samples compared to non-tumor samples in CD4^+^ (*p* < 0.05), CD8^+^ (*p* < 0.001) and CD4^+^ CD8^+^ (*p* < 0.05) T cells, and a decrease on HLA-DR^+^ T-cells, being significant in CD4^+^ (*p* < 0.001), CD8^+^ (*p* < 0.001), and CD4^+^ CD8^+^ (*p* < 0.05) T cells (Figure 5).

## 4. Discussion

Liver metastasis remains the principal cause of death in CRC patients, in which its formation decreases up to 70% of the overall five-year survival rate compared to patients with localized disease [4,5]. Unquestionably, the immune system has significant importance in tumor outgrowth and metastasis formation. It has been a focus of study to design new personalized therapies to provide a better outcome for CRC patients [27]. 

This study entails a thorough analysis and identification of 60 different T-cell populations present in human samples of CRC liver metastasis using flow cytometry analysis due to its elevated multiparametric capability and increased results reproducibility.

The infiltrated T-lymphocytes have been described as essential cells in CRC primary tumors and liver metastasis [16,28]. We found a decreased density of these cells in the tumor samples compared to the non-tumor, even when we removed from the total cellularity the non-hematopoietic cells. The results also revealed a decreased percentage of CD8^+^ T cells, increased CD4^+^ T cells, and a significant decrease in the frequency of CD4^+^ and CD8^+^ expressing CD56 (NK-like phenotype), which indicates a diminished migration of cells with cytotoxic ability into the TME leading to the tumor cells proliferation and outgrowth. 

Furthermore, the tumor samples were subdivided according to their histopathological growth pattern into desmoplastic and non-desmoplastic. The growth pattern describes the morphology and interaction between tumor and liver cells at the tumor-liver interface [7]. The desmoplastic pattern presented a higher percentage of CD8^+^ T cells compared to the non-desmoplastic pattern. In contrast, the density of CD4^+^ T cells decreased in the same conditions. In this regard, it has been described that the CD8^+^ T cells play a crucial role in metastatic suppression and are associated with a decrease in distant metastasis formation, improving the outcome for CRC patients [6,29]. This increased cytotoxicity was also verified by a higher frequency of CD4^+^ and CD8^+^ T cells expressing CD56 which characterizes an NK-like phenotype, pointing to an improved outcome for patients with tumors with desmoplastic growth patterns compared to non-desmoplastic ones. 

Since the immune system is crucial in developing malignancies, treating cancers like CRC may benefit from this new therapeutic approach [30]. Tregs are essential for controlling homeostasis and antitumor immune responses, and their infiltration is linked to weakened antitumor cell responses [15,31]. Additionally, these cells modify their transcriptional program in response to the numerous cytokines they are exposed to in the inflammatory environment. Transcription factors typically linked to the development of other effectors CD4^+^ T-cell subsets regulate this adaptability. Salama et al. associated a higher density of Treg cells with a better outcome for CRC patients leading to the hypothesis that Treg-mediated suppression of chronic inflammation may prevail over the control of antitumor immunity [32]. However, other data strongly suggest that Tregs do not inhibit but rather promote pro-tumor Th17 responses and actively suppress tumor-specific CD8^+^ T cell activation in CRC patients [33]. Our results showed that Tregs density in tumor samples of CRC liver metastasis is increased compared to non-tumor samples. 

The activation phenotype of CD4^+^ Tregs was evaluated through the frequency of HLA-DR. This marker is expressed during mid-to-late periods after T-cell activation [34]. HLA-DR is an MHC class II molecule displayed on the T-cell surface upon activation, which can trigger some activation signals and, in turn, allows those cells to present processed antigenic peptides [35]. This increased presence of Treg HLA-DR^+^ cells indicates that the Treg functions are stimulated, increasing the evidence that these cells might be a key target to inhibit CRC liver metastasis. The depletion of Treg cells has also been linked to an increase in anti-tumor immune responses and a decrease in tumor burden [15,31]. Although Treg cell-depleting techniques have produced encouraging clinical outcomes, some pertinent difficulties need to be solved to make them safer, more efficient, and more widely used in treating CRC hepatic metastases [36]. 

Overall, combined with the diminished density of CD8^+^ T cells within the TME, we can conclude that Treg cells contribute to an immunosuppressive microenvironment, which promotes the cancer cells‘ proliferation and progression. 

Follicular T cells play a crucial role in the development of humoral immunity by regulating the cellular reactions occurring in the germinal center [22,37]. These cells are also considered critical mediators in the immune pathogenesis of CRC [22]. CD185 is the most common marker for the follicular-like phenotype evaluation on T cells. The most described CD185-expressing T cells are CD4^+^, designed as follicular helper T cells [38]. Recently, CD8^+^ CD185^+^ have also been studied in several pathologies, such as CRC, named follicular cytotoxic T cells [39]. 

In this study, we demonstrated that the TME in CRC liver metastasis does not influence the density of the CD4^+^ CD185^+^ cells, even decreasing its percentage when activated (HLA-DR^+^). The frequency of CD25^+^ cells was also assessed to define the activation profile of these cells. CD25 is commonly related to Treg cells, which is a well-known surface marker and is described as a potential target to inhibit CRC progression [40]. Additionally, CD25 is an intermediate activation marker, classified as an α-chain of the IL-2 receptor complex that is constitutively present on a subset of peripheral blood lymphocytes with antigen density increasing in vitro upon activation by stimulation of the TCR/CD3 complex [40,41]. An extremely low density of CD25^+^ follicular-like T cells in all the subpopulations used in this analysis was detected. In contrast, CD8^+^ CD185^+^ cells were significantly increased in the tumor samples, indicating increased cytotoxicity within the tumor, in discrepancy with our results. 

CD8^+^ CD185^+^ T cells arise and respond to cancer cells, suggesting a prevalent role of chronic antigen exposure in these T cells’ development [37]. Further, CD8^+^ CD185^+^ T cells in CRC maintain a cytolytic capacity to directly lyse tumor cells but can also influence B cell secretion of IgG, suggesting multiple mechanisms for tumor control by these cells [42]. Despite the rich cytolytic potential and activity by CD8^+^ CD185^+^ T cells, tumor cells likely employ inhibitory mechanisms to suppress these immune cells’ function. Additionally, Chu et al. have demonstrated the importance of CD185, in which CD8^+^ CD185^+^ T cells exert superior antitumor effects, including increased proliferative/migratory potential, secretion of effector molecules, and suppression of tumor-promoting T helper activities compared to CD8^+^ CD185^−^ T cells [43]. Furthermore, pancreatic cancer and CRC disease-free survival time are both positively correlated with CD8^+^ CD185^+^ T cell frequency [37,44]. Taking into consideration this knowledge and our results, the presence of a follicular-like phenotype in cytotoxic T cells may extend cancer treatment efficacy.

We also evaluated this follicular-like phenotype in the other major T-cell populations, showing a significant increase in CD4^−^ CD8^−^ CD185^+^ T cells. To our knowledge, this phenotype has exclusively been studied in CD4^+^ or CD8^+^ cells.

T helper lymphocytes are considered to be the prominent players in tumor immunity and, especially, to influence tumor progression [18]. Th1 cells arise from naïve T cells through stimulation by interleukin-12 (IL-12) and produce interferon-gamma (IFN-γ), IL-2, and tumor necrosis factor-α (TNF-α). Th1 cells induce B cells to release antibodies of the immunoglobulin G (IgG) isotype. These antibodies are responsible for phagocyte activation and antibody-dependent cellular cytotoxicity, supporting the cytotoxic T lymphocyte response [45]. These cells are characterized by the expression of CCR5 and/or CXCR3 surface markers [46]. IL-6 and transforming growth factor (TGF)-β activation promote the development of Th17 cells. Th17 cells are the primary IL-17 producers and actively contribute to the development of autoimmune diseases, tumors, and inflammation [47]. CCR6 is the surface marker used for the identification of these cells. Moreover, several studies demonstrate that a high density of Th17 cells predicts a poor prognosis in CRC patients [17,48]. Additionally, in CRC, multiple studies also indicate that IL-17 influences tumor initiation and progression [17]. Here, we demonstrated that tumor samples of CRC liver metastasis have a significantly increased density of Th17 cells compared to non-tumor samples. We also obtained a decreased frequency of CCR5^+^ CXCR3^−^ cells that we considered as Th1 cells. Th1 phenotype was characterized by the expression of at least one well-established surface marker [49]. CD4^+^ cells with no frequency of CCR5, CXCR3, and CCR6, can be any other functional T helper cells, such as Th2, Th9, Th22, or Th0. Additionally, we also verify a decreased presence of Th1/Th17 cells that express CCR6 but also express at least one well-established Th1 surface marker, indicating a plasticity phenotype of Th17 towards a Th1 phenotype. 

In CD8^+^ T cells, this phenotype was also evaluated, also indicating an increased presence of Tc17, as well as, Tc17/Tc1 cells, and a diminished density of Tc1 cells. This suggests an overall increase of IL-17-producing cells in the TME that should contribute to a significantly increased expression of IL-17 in the liver niche [50]. The IL-17 expression may be one of the potential biomarkers for developing a new prognostic target for CRC. Despite the fact that this interleukin has been studied in primary CRC, further studies are needed to evaluate its expression on the metastatic site [51]. Additionally, CXCR3 has also been linked to the acquisition of an effector or memory precursor phenotype [52]. Duckworth et al. have described that in the absence of CXCR3, the T cells are confined to the lymph node center and alternatively differentiate into stem-like memory cell precursors [53]. Our results revealed an increased presence of CD8^+^ CCR6^+^ CCR5^−^ CXCR3^+^, which can be correlated with the acquisition of an effector phenotype and further antitumor responses contributing to the immune system‘s action against cancer cells [53,54]. Although CXCR3 has been characterized in viral infections, further studies are needed to evaluate this cytokine’s role in metastatic cancer formation, especially in CRC. Taking this knowledge into account, CCR5^−^ CXCR3^+^ cells could be a potential target for new CRC therapies.

To corroborate these results, we also assessed the density of the T cells with the initial tumor size. As the tumor size increases, the presence of CD4^+^ CCR6^+^ CCR5^+^ CXCR3^−^ and CD4^+^ CCR6^+^ CCR5^−^ CXCR3^−^ T cells also increase, which means that Th17/Th1 and Th17 cells’ presence is directly correlated with the size of the tumor in CRC liver metastasis. These additional results reinforce the need for novel studies to establish IL-17-secreting cells as targets for new personalized therapies.

Finally, the activation profile of these functional T-cell subsets was also evaluated. There is no doubt that the heterogeneity in cell activation and turnover among the CD4^+^ or CD8^+^ T-cell populations is tightly linked to their differentiation status and their interplay with antigen stimulation and cell homeostasis [35]. Several markers are used to evaluate the activation profile of T cells, including early, intermediate, and late activation markers [55]. In this study, the activation profile of the T cells’ major subpopulations was accessed by the frequency of both CD25 and HLA-DR, already used in our previous analysis. 

The T cells’ major populations, CD4^+^, CD8^+^, CD4^+^ CD8^+^, CD4^−^, CD8^−^, and γδ^+^ cells, have increased levels of CD25 and decreased levels of HLA-DR positive cells, except for γδ^+^ T cells. Nowicka et al. have described that this intermediate activation marker was substantially increased in peripheral blood from ovarian cancer patients compared to its expression in the tumor-infiltrating lymphocytes [56]. They also demonstrated that the frequency of HLA-DR^+^ T cells increased in the tumor-infiltrated lymphocytes compared to the peripheral blood. HLA-DR expression in tumors has been positively associated with enhanced lymphocytic infiltration. HLA-DR has been used as a surrogate marker of immune competence in various clinical studies, and its loss in circulation has been linked with susceptibility to post-surgical infection and sepsis [57]. According to our analysis, tumor samples have less T-lymphocyte infiltration compared to the non-tumor samples representing a possible explanation for the decreased frequency of HLA-DR^+^ T-cell subsets. 

Note that in this study, the CD25 expression was evaluated after removing the Treg cells percentage to guarantee that we were only accessing the activation profile. In terms of activation, and considering Nowicka et al.’s results, cells in the metastatic site might maintain the characteristics gained by their dislocation in the circulatory system [56]. The increased expression of this intermediate activation marker in the liver niche may indicate that the micro-environmental conditions are not prone to an increased late activation profile. This can suggest that CD25 might be a possible target to inhibit CRC liver metastasis progression.

## 5. Conclusions

In summary, this study revealed that the immune microenvironment within CRC liver metastasis lacks infiltrated lymphocytes and presents an immunosuppressive profile compared to the non-tumor samples. It also revealed an increased density of Treg cells with and without an activated phenotype, and an increased presence of IL-17-releasing cells in the TME, potentiating the acquisition of an immunosuppressive phenotype. In contrast, CD8^+^ CD185^+^ cells and effector CD8^+^ T cells were increased, enhancing the antitumor activities targeting and promoting cancer cell lysis. CD25 can also be a new target for CRC liver metastasis inhibitions due to its increased frequency in the T-cell subpopulations within the tumor. 

With this study, we take a step further in the characterization of CRC liver metastasis, but additional research is crucial to understand the molecular mechanisms involved in the CRC metastatic cascade and create specific targets to improve the patient’s outcome. These new targets and prognostic biomarkers are an advance in new precision medicine approaches that, from our point of view, are the future of CRC treatment.

## Figures and Tables

**Figure 1 cancers-14-06069-f001:**
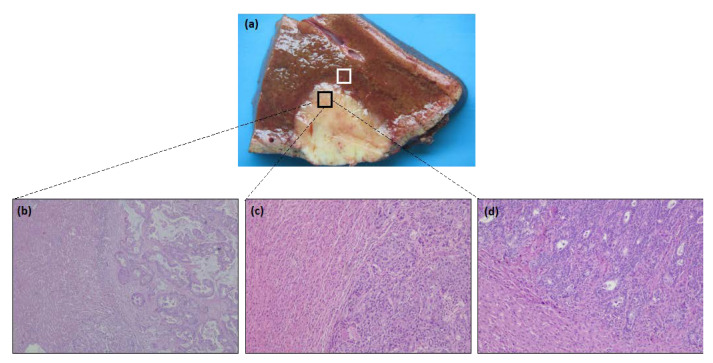
Liver gross section with colorectal metastases and representative histological images of the growth pattern of liver metastasis of CRC. (**a**) Tissue was collected at the tumor/non-tumoral interface, according to the picture—tumor (black) and non-tumoral (white). (**b**) Representative images of the immunohistochemistry-stained tumor sections with desmoplastic growth pattern (H&E 20×), (**c**) expansive growth pattern (H&E 40×), and (**d**) replacement growth pattern (H&E 40×).

**Figure 2 cancers-14-06069-f002:**
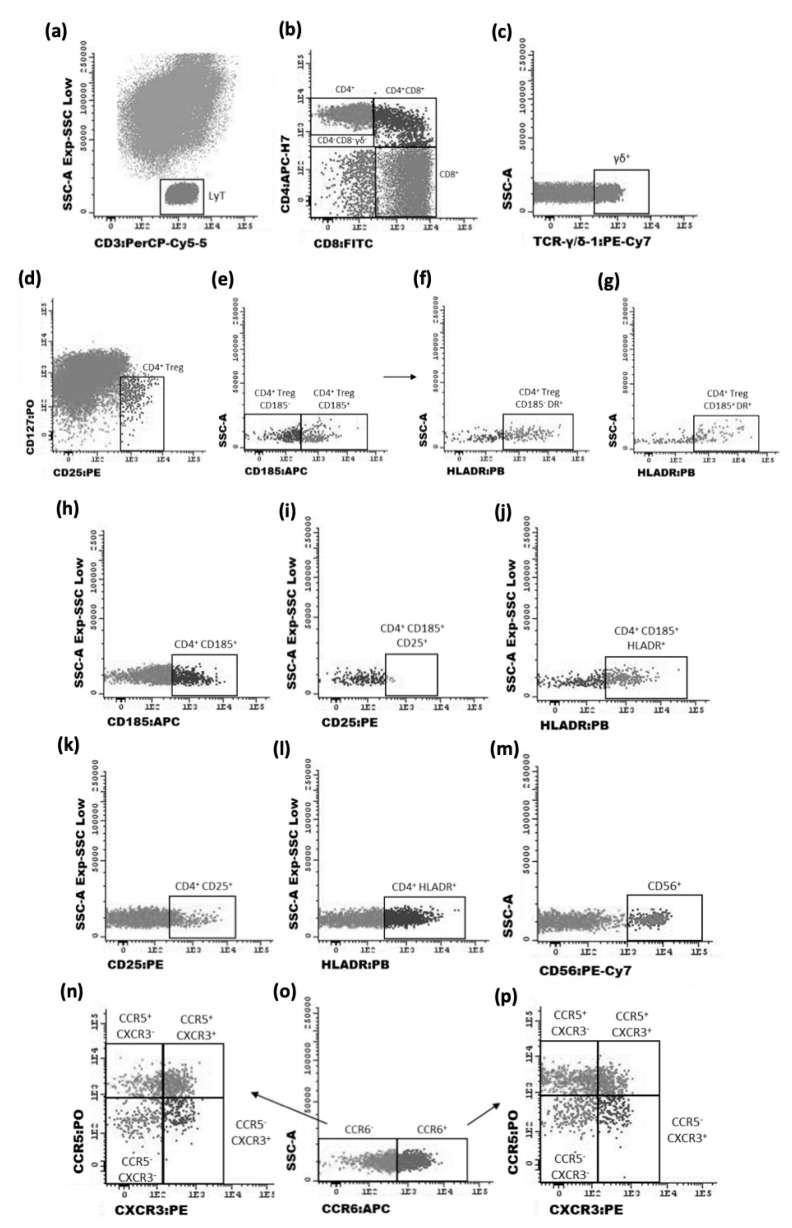
Representative dot plot histograms showing the gating strategy used for the identification of CD4 T-cell subpopulations. T lymphocytes were identified by the expression of CD3 (**a**); among them, five major subpopulations were found (CD4^+^, CD8^+^, CD4^+^ CD8^+^, CD4^−^CD8^−^and γδ^+^ T cells) (**b**,**c**). Treg cells were identified as CD25^bright^ CD127^−/low^ (**d**) and the subpopulation follicular-like Tregs by the expression of CD185 (**e**), as well as its activation profile by the expression of HLA-DR^+^ (**f**,**g**). The follicular-like T-cell subsets were evaluated by CD185 expression and its activation profile (CD25^+^ or HLA-DR^+^) (**h**–**j**). The activated (CD25^+^ or HLA-DR^+^) non-follicular non-Treg CD4 T cells were also measured (**k**,**l**). The percentage of CD4 T cells with an NK-like phenotype (CD56^+^) was also evaluated (**m**). From the functional point of view, CD4 T cells were characterized as CCR5^+^ CXCR3^+^, CCR5^+^ CXCR3^−^, CCR5^−^ CXCR3^+^, CCR5^−^ CXCR3^−^, CCR6^+^, CCR6^+^ CCR5^+^ CXCR3^+^, CCR6^+^ CCR5^+^ CXCR3^−^, CCR6^+^ CCR5^−^ CXCR3^+^, and lastly, CCR6^+^ CCR5^−^ CXCR3^−^ (**n**–**p**). This gating strategy was repeated for the other major T cell populations: CD8^+^, CD4^+^, CD8^+^, CD4^−^ CD8^−^ and γδ^+^ cells. However, the described functional characterization, as well as the CD56 expression, was only evaluated in CD4^+^ and CD8^+^ T cells due to the much lower percentage of cells corresponding to the other T cell populations.

**Figure 3 cancers-14-06069-f003:**
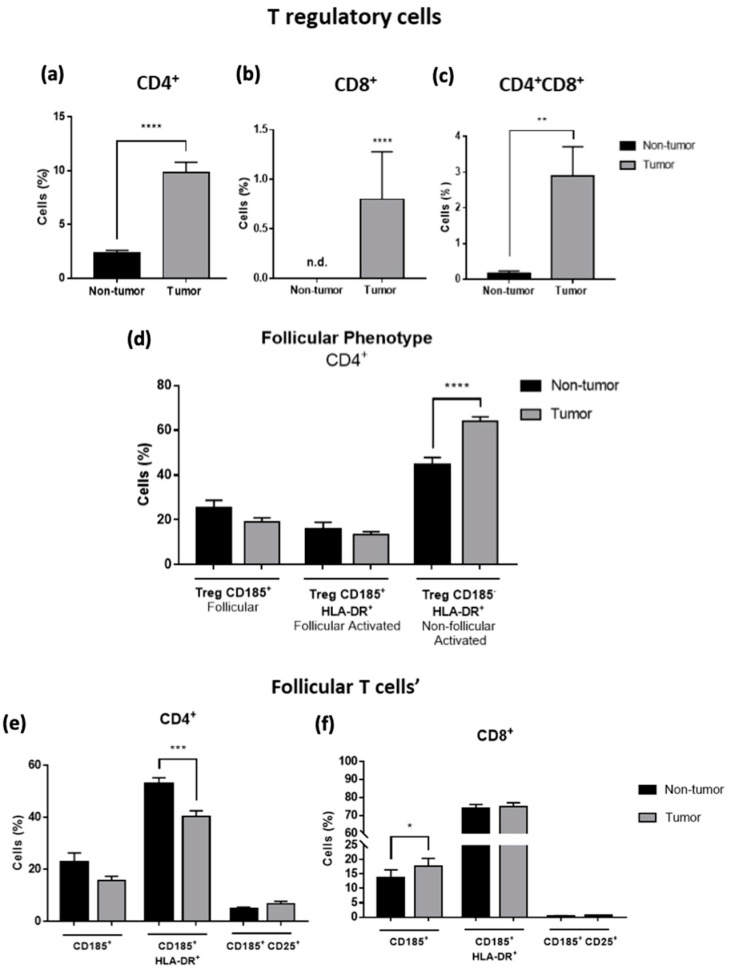
Flow cytometry analysis of T cells populations present in non-tumor and tumor samples of CRC liver metastasis. Percentage of Tregs in CD4^+^ (**a**), CD8^+^ (**b**) and CD4^+^CD8^+^ (**c**) T lymphocytes. (**d**) Percentage of CD4^+^ CD185^+^ Tregs (follicular phenotype), and activated by the expression of the HLA-DR marker. (**e**–**i**) Analysis of the other follicular T cells (CD4^+^, CD8^+^, CD4^+^CD8^+^, CD4^—^CD8^—^ and γδ^+^ T lymphocytes), as well as their activation status by the expression of CD25 or HLA-DR. All results are shown as mean ± SEM. * *p* < 0.05, ** *p* < 0.01, *** *p* < 0.001 and **** *p* < 0.0001 were significantly different when compared to non-tumor samples using Mann–Whitney tests.

**Figure 4 cancers-14-06069-f004:**
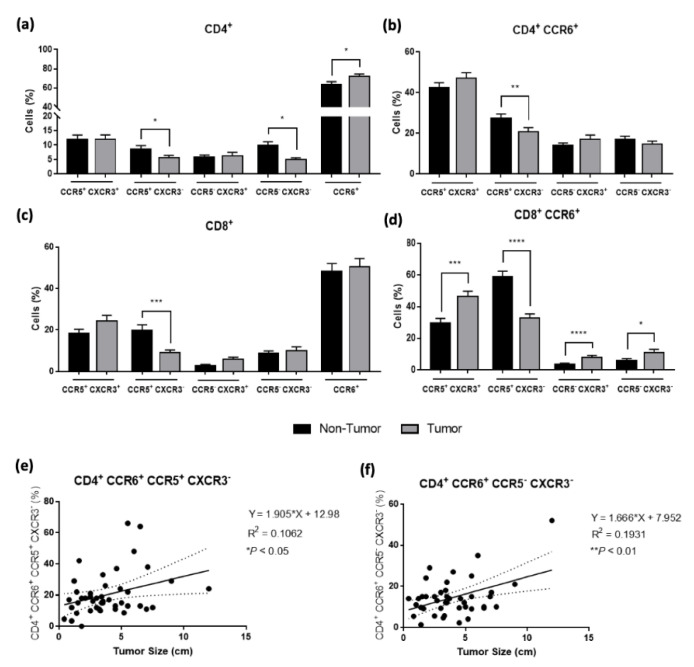
Flow cytometry analysis of T-cell functional phenotypes. Frequencies of the different functional CD4^+^ (**a**) or CD8^+^ (**c**) T-cell subsets based on the expression of CCR5, CXCR3, and CCR6. (**b**) Analysis of CD4^+^ CCR6^+^ and (**d**) CD8^+^ CCR6^+^ T-cell subsets. All results are presented as mean ± SEM. * *p* < 0.05, ** *p* < 0.01, *** *p* < 0.001 and **** *p* < 0.0001 were significantly different when compared to non-tumor samples using Mann–Whitney tests. (**e**,**f**) Correlation between the percentage of CD4^+^ CCR6^+^ CCR5^+^ CXCR3^—^ and CD4^+^ CCR6^+^ CCR5^—^ CXCR3^—^T cells, respectively, with the tumor size (cm), confirmed by a Pearson test.

**Figure 5 cancers-14-06069-f005:**
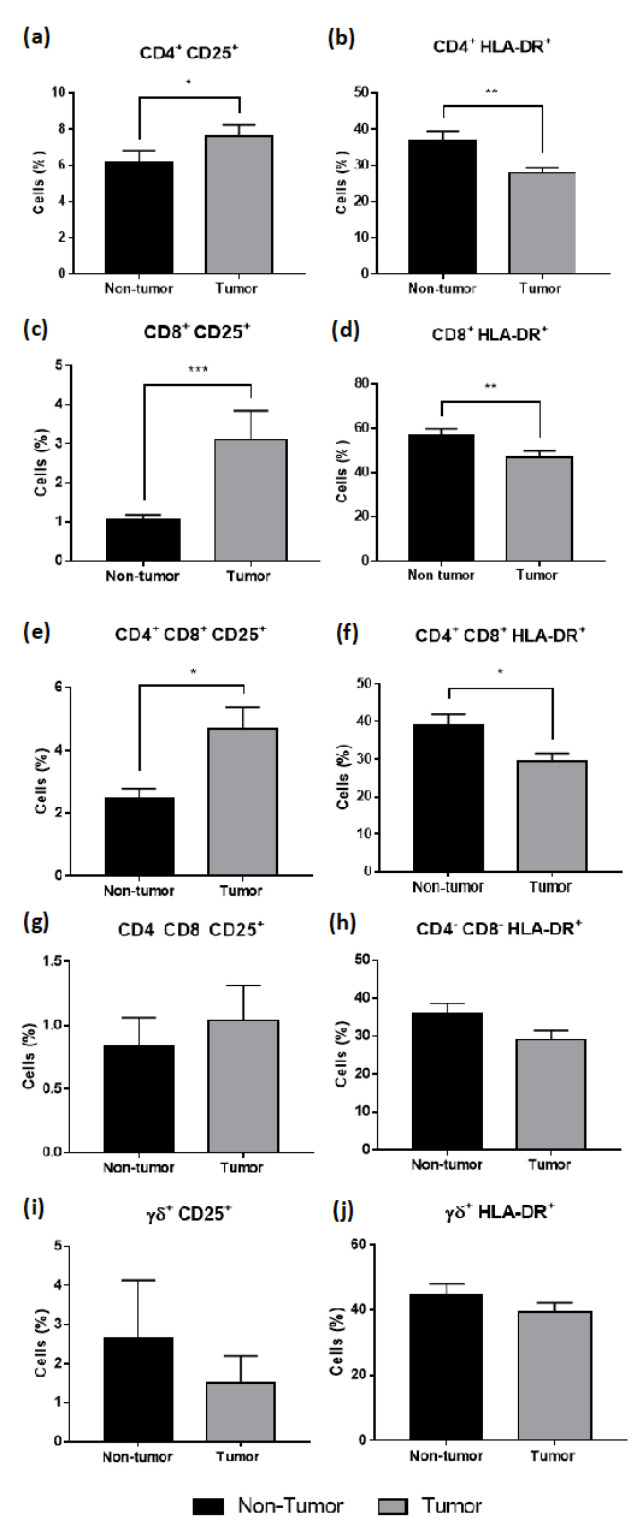
Analysis of the activation profile of T cells in non-tumor and tumor samples of CRC liver metastasis. Assessment of the expression of CD25^+^ and HLA-DR^+^ in CD4^+^ (**a**,**b**), CD8^+^ (**c**,**d**), CD4^+^CD8^+^ (**e**,**f**), CD4^—^CD8^—^ (**g**,**h**), and γδ^+^ (**i**,**j**). All results are shown as mean ± SEM. * *p* < 0.05, ** *p* < 0.01 and *** *p* < 0.001 were significantly different when compared to non-tumor samples using Mann–Whitney tests.

**Table 1 cancers-14-06069-t001:** Demographic and clinical characteristics.

Variable	(*n* = 47)
**Age at surgical resection**	
Mean ± SD; range	59.1 ± 12.8; 36–90
**Variable**	**Number (%)**
**Gender**	
Male	31 (66)
Female	16 (34)
**Presentation**	
Synchronous	26 (55)
Metachronous	21 (45)
**Primary tumor location**	
Sigmoid	16 (34)
Ascending	10 (21)
Descending	5 (11)
Splenic Flexure	4 (9)
Hepatic Flexure	1 (2)
Transverse	2 (4)
Rectal	9 (19)
**T stage of the primary colorectal tumor**	
3	31 (66)
4a	12 (26)
4b	2 (4)
**First approach**	
Liver First	4 (9)
Colectomy	34 (72)
Synchronous resection	7 (15)
**Preoperative systemic chemotherapy**	27 (57)
**Variable**	**(*n* = 47)**
**Number of colorectal liver metastases**	
Mean ± SD; range	2.3 ± 1.8; 1–10
**Size of the largest colorectal liver metastases (cm)**	
Mean ± SD; range	3.8 ± 2.4; 0.4–12
**Variable**	**Number (%)**
**Histologic growth pattern**	
Desmoplastic	17 (36)
Non-desmoplastic	30 (64)
**Status**	
Dead	3 (6)
Alive	44 (94)

**Table 2 cancers-14-06069-t002:** The panel of monoclonal antibodies used for the T cells’ phenotypic characterization, indicating their respective volume, the fluorochrome, commercial source, and clone.

Fluorochromes								
Tube	PB	PO	FITC	PE	PerCP-Cy5.5	PE-Cy7	APC	APC-H7
1	HLA-DR	CD127	CD8	CD25	CD3	TCR-γ/δ-1	CD185	CD4
	BD (L243)	BD Horizon (HIL-7R-M21)	BD (SK1)	BD (2A3)	BD (SK7)	BD (11F2)	R&D Systems (51505)	BD (SK3)
	2 μL	5 μL	10 μL	10 μL	10 μL	1 μL	5 μL	5 μL
2	CD4	CCR5	TCR-γ/δ-1	CXCR3	CD3	CD56	CCR6	CD8
	BD Horizon (RPA-T4)	BD OptiBuild (2D7/CCR5)	BD (11F2)	BD Pharmingen 1C6/CXCR3	BD (SK7)	Beckman Coulter N901 (NKH-1)	BD Pharmingen (11A9)	BD (SK1)
	2 μL	5 μL	7 μL	15 μL	10 μL	2.5 μL	5 μL	2.5 μL

Abbreviations: APC—allophycocyanin; APC-H7—allophycocyanin-hilite 7; FITC—fluorescein isothiocyanate; PB—pacific blue; PE—phycoerythrin; PE-Cy7—phycoerythrin-cyanine 7; PerCP-Cy5.5—peridinin chlorophyll protein-cyanine 5.5; PO—pacific orange. Commercial sources: BD (Becton Dickinson Biosciences, San Jose, CA, USA); BD Pharmingen (San Diego, CA, USA); BD Horizon (Franklin Lakes, NJ, USA); Beckman Coulter (Miami, FL, USA); R&D Systems, Inc. (Minneapolis, MN, USA); BD OptiBuild (San Diego, CA, USA).

**Table 3 cancers-14-06069-t003:** Percentage of T cells in non-tumor and tumor samples of liver metastasis of CRC with non-desmoplastic and desmoplastic growth pattern.

Cell Types	Non-Tumor (%) ± SEM	Tumor (%) ± SEM	Tumor	
			Non-Desmoplastic (%) ± SEM	Desmoplastic (%) ± SEM
T cells	21.75 ± 2.29	6.78 ± 1.30 ^d^	5.40 ± 1.17	9.21 ± 2.90
T cells (after removing the non-hematopoietic cells)	30.21 ± 3.18	21.87 ± 4.18 ^b^	24.30 ± 3.35	34.83 ± 5.88
CD4^+^	21.71 ± 1.46	40.54 ± 2.72 ^d^	40.12 ± 3.66	41.29 ± 3.96
CD56^+^	8.14 ± 0.83	2.07 ± 0.27 ^d^	1.43 ± 0.18	3.09 ± 0.56 **
CD8^+^	60.71 ± 2.10	39.35 ± 2.51 ^d^	35.16 ± 3.09	46.75 ± 3.78 *
CD56^+^	34.69 ± 2.22	16.04 ± 1.49 ^d^	14.91 ± 1.67	17.90 ± 2.84
CD4^+^ CD8^+^	4.00 ± 0.39	2.21 ± 0.28 ^d^	1.93 ± 0.28	2.69 ± 0.60
CD4^−^ CD8^−^	7.59 ± 1.45	13.16 ± 3.22	17.14 ± 4.54	6.13 ± 3.39 *
γδ ^+^	5.90 ± 0.68	2.76 ± 0.55 ^d^	2.52 ± 0.77	3.18 ± 0.73

^b^*p* < 0.01 and ^d^
*p <* 0.0001 significantly different when compared to non-tumor samples, using Mann–Whitney non-parametric tests; * *p* < 0.05 and ** *p* < 0.01 significantly different when compared to tumor samples with a non-desmoplastic growth pattern also using Mann–Whitney non-parametric tests. All results are shown as mean ± SEM.

## Data Availability

Not applicable.

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
