# Peer review of "Extensive Phenotypic Characterization of T Cells Infiltrating Liver Metastasis from Colorectal Cancer: A Potential Role in Precision Medicine"

_cancers, 2022, doi:10.3390/cancers14246069_

Round 1
Reviewer 1 Report
Very interesting work. A further developing work maybe - what about exhausted lymphocytes - identification and characteristics ?
Author Response
We thank the reviewer for the comment. Please see the attachment for our response.

Reviewer 2 Report
PEER REVIEW: Extensive phenotypic characterization of T cells infiltrating liver metastasis from colorectal cancer: a potential role in precision medicine
Major comments
- -In the introduction, there's no mention of the role of follicular T lymphocytes in tumors, and it's a population that is widely studied in your research. I think that a comprehensive explanation of the impact of this population's modification in tumors could give the lector a better comprehension.
- -In your results, you showed, according to different markers, different populations of Th1 and Th17 cells. However, I think it is crucial to evaluate these T lymphocyte cytokine production, analyzing INF-γ and IL-17 production, respectively, to corroborate the presence and functionality of these cells.
- -I consider it essential to mention, either in material and methods or as supplemental material, the different controls used in the flow cytometry analysis, including compensation controls, fluorescence minus one (FMO) control, etc.
- -Currently, different tools are available for the analysis of high-parameter flow cytometry intuitively, such as t-SNE (t-Stochastic Neighbor Embedding) or FlowSOM. Why didn't you use this alternative to evaluate your results for a better interpretation and minimize errors?
Minor comments
-It could be interesting to know, apart from the difference between desmoplastic and non-desmoplastic tumors, if in desmoplastic tumors are any differences between the different patterns of desmoplastic tumors (immature, intermediate, and mature), since where according to the literature, there are further T lymphocyte distribution according to the desmoplastic pattern.
-Some patients were on a chemotherapy regimen at the moment of tissue recollection. It is known that chemotherapy can influence blood cell lines. Can chemotherapy affect the presence of lymphocytes in tumor samples?
Author Response
We thank the reviewer for the comments and suggestions. Please see the attachment for our response.

Reviewer 3 Report
This is certainly an appropriate study and very timely. The methodology and interpretation of the results, based on the current cancer immunotherapy knowledge is fitting.
In future studies the authors need to address the contribution of the microbiota that has been shown to cause dysbiosis often leading to initiation, maintenance, and metastasis.
CRC is even more common in developing countries where diagnosis is often late and prognosis is poor. How can this personalized patient directed approach be made to have maximum global impact. Cited, are the potential biomarker targets for diagnostics and prognostics. This is a step in the right direction.
Author Response
We thank the reviewer for the comment and suggestions. Please see the attachment for our response.

Reviewer 4 Report
Review Report for the Manuscript “Extensive phenotypic characterization of T cells infiltrating liver metastasis from colorectal cancer: a potential role in precision medicine”
Rating the Manuscript
Originality/Novelty: Is the question original and well defined? Do the results provide an advance in current knowledge?
Yes, using flowcytometry studies, the authors have studied the phenotypic characterization of T cells infiltrating liver metastasis from colorectal cancer which could be helpful in precision medicine.
Significance: Are the results interpreted appropriately? Are they significant? Are all conclusions justified and supported by the results? Are hypotheses and speculations carefully identified as such?
Yes, the results are interpreted well, and the conclusions are justified by the results.
Quality of Presentation: Is the article written in an appropriate way? Are the data and analyses presented appropriately? Are the highest standards for presentation of the results used?
Yes, the article is written well.
Scientific Soundness: is the study correctly designed and technically sound? Are the analyses performed with the highest technical standards? Are the data robust enough to draw the conclusions? Are the methods, tools, software, and reagents described with sufficient details to allow another researcher to reproduce the results?
Yes, the data is robust enough to draw conclusions and the methods, tools and software used in the data analysis are explained properly.
Interest to the Readers: Are the conclusions interesting for the readership of the Journal? Will the paper attract a wide readership, or be of interest only to a limited number of people? (Please see the Aims and Scope of the journal)
Yes, this would be a great article for the researchers in the cancer research field.
Overall Merit: Is there an overall benefit to publishing this work? Does the work provide an advance towards the current knowledge? Do the authors have addressed an important longstanding question with smart experiments?
Yes. This study provides an advancement to the current knowledge.
English Level: Is the English language appropriate and understandable?
Yes, English language in the manuscript is appropriate and understandable.
Overall Recommendation: Accept after Minor Revisions
The manuscript is written well and there are only few comments. Given below are the comments for each section of the manuscript.
Author list:
Line 5: Gabriela Sampaio-Ribeiro1,2,3,†, Ana Ruivo4,5, † , Ana Silva1, Ana Lúcia Santos1, Rui Caetano Oliveira2,6,7,8, Paula Laranjeira1,2,3,9, João Gama10, Maria Augusta Cipriano10, José Guilherme Tralhão2,4,5,7,8, ††, Artur Paiva1,2,3,11, * and ††
† These authors contributed equally to this work
†† These authors contributed equally to this work
What there are two sets of names stating that two sets of authors are equally contributed to the manuscript.
Also as highlighted at the end there’s the word “and”. Check and correct the error accordingly.
Simple Summary and Abstract
Simple summary and the abstract well summarizes the content of the manuscript.
Simple Summary:
Line 30: “In this study, we characterized several T cell populations by flow cytometry in non-tumor and tumor samples of CRC liver metastasis, both subdivided according to their growth pattern into desmoplastic and non-desmoplastic”
It’s better if the authors could briefly explain what types of T cell populations they characterized in this study.
Introduction
Introduction is well written.
Line 98: “Taking this into account, the purpose of this study was to identify around 60 different T cells subpopulations in tissue samples from patients with CRC liver metastasis, which are further subdivided into non-tumor and tumor samples with non-desmoplastic or desmoplastic growth patterns, in order to characterize the TME and clarify new potential targets to inhibit CRC liver metastasis progression.”
At the end of the introduction authors could mention what methods are used in the study.
Materials and Methods:
Methods section is very well written with all the required information.
Figures and Tables:
Figure 1: If possible, include scale bars for figure 1 b, c, and d.
All the other figures and tables are good.
Discussion
Line 342: “This study entails a thorough analysis and identification of 60 different T cell populations present in human samples of CRC liver metastasis using flow cytometry analysis.”
What are the other methods that could be used for identification of different T cell populations? Why you selected flowcytometry and what are the advantages of using flowcytometry?
References:
Some of the references are more than 10 years old. It they don’t contain important information authors could replace these with new references.
References: 5,13,18,20,21,24,25,29,31,33,39,45,46,53 and 54

Author Response

(The authors gave the same response as above.)
